# Mitochondrial Genetic and Epigenetic Regulations in Cancer: Therapeutic Potential

**DOI:** 10.3390/ijms23147897

**Published:** 2022-07-18

**Authors:** Alexandra Wagner, Helena Kosnacova, Miroslav Chovanec, Dana Jurkovicova

**Affiliations:** 1Department of Genetics, Cancer Research Institute, Biomedical Research Center, Slovak Academy of Sciences, 845 05 Bratislava, Slovakia; alexandra.wagner@savba.sk (A.W.); helena.svobodova@savba.sk (H.K.); miroslav.chovanec@savba.sk (M.C.); 2Department of Simulation and Virtual Medical Education, Faculty of Medicine, Comenius University, 811 08 Bratislava, Slovakia

**Keywords:** cancer, mitochondria, genetics, epigenetics, DNA repair, mitomiRs, targeted therapy

## Abstract

Mitochondria are dynamic organelles managing crucial processes of cellular metabolism and bioenergetics. Enabling rapid cellular adaptation to altered endogenous and exogenous environments, mitochondria play an important role in many pathophysiological states, including cancer. Being under the control of mitochondrial and nuclear DNA (mtDNA and nDNA), mitochondria adjust their activity and biogenesis to cell demands. In cancer, numerous mutations in mtDNA have been detected, which do not inactivate mitochondrial functions but rather alter energy metabolism to support cancer cell growth. Increasing evidence suggests that mtDNA mutations, mtDNA epigenetics and miRNA regulations dynamically modify signalling pathways in an altered microenvironment, resulting in cancer initiation and progression and aberrant therapy response. In this review, we discuss mitochondria as organelles importantly involved in tumorigenesis and anti-cancer therapy response. Tumour treatment unresponsiveness still represents a serious drawback in current drug therapies. Therefore, studying aspects related to genetic and epigenetic control of mitochondria can open a new field for understanding cancer therapy response. The urgency of finding new therapeutic regimens with better treatment outcomes underlines the targeting of mitochondria as a suitable candidate with new therapeutic potential. Understanding the role of mitochondria and their regulation in cancer development, progression and treatment is essential for the development of new safe and effective mitochondria-based therapeutic regimens.

## 1. Introduction

Cancer cells display distinct molecular, morphological and functional features when compared to normal cells. Malignant transformation of the cell is a multistep process that starts with genetic mutations and involves selective growth, metabolic switching, proliferative advantage and altered stress response, resulting in pathological vascularization, invasion and metastasis [1]. Mitochondria, which arose through evolution presumably as a proteobacteria swallowed by a pre-eukaryotic cell, are organelles with their own DNA encoding proteins that regulate numerous metabolic processes and signalling pathways involved in most cellular functions [2,3]. Detailed characterizations of mitochondria functioning in physiological environments have comprehensively been described in numerous reviews [4,5,6]. A relationship between mitochondrial bioenergetics and cancer was first proposed in 1920s, when Otto Warburg demonstrated that cancer cells increase glucose uptake and use glycolysis as a main source for ATP production even in the presence of oxygen, this being attributed to the metabolic impairment of oxidative phosphorylation [7,8]. In cancer cells, glycolysis is the favoured metabolic pathway. However, oxidative phosphorylation (OXPHOS) may just be downregulated or exhibit standard activity [9,10,11]. In addition to the strong dependence on glycolysis, many tumours or subpopulations of cells in tumours rely on OXPHOS for bioenergetics [12,13,14] and biosynthetic processes [15,16], which provide the basis for the therapeutic targeting of mitochondria. Furthermore, an increased dependence of tumour cells on OXPHOS was observed in the advanced stages of the disease [17]. During tumorigenesis, various signalling cascades regulating mitochondrial metabolism are altered [18], reflecting the ability of mitochondria to adjust the metabolic demands of cancer cells towards aberrant survival. In this regard, mitochondria should be viewed as organelles crucial not only for tumorigenesis and cancer development and progression, but also highly important for treatment response in cancer cells. In addition, genetic and epigenetic features of mitochondria and their bi-directional communication with the nucleus highlight their importance in cancer therapy response. Numerous cancer treatments based on the use of DNA damaging agents, affecting nuclear DNA (nDNA) as well as mitochondrial DNA (mtDNA), are expected to trigger apoptosis and eliminate deregulated cancer cells. In the nucleus, an aberrant capacity for homologous recombination (HR) and nucleotide excision repair (NER) critically contributes to avoidance of apoptosis and leads to drug resistance; in mitochondria, mechanisms of DNA damage response (DDR) and repair still require more elucidation. However, targeting DDR and repair pathways in both the nucleus and mitochondria, together with modulation of epigenetics and miRNA regulation, remains a plausible and efficient strategy to overcome drug resistance. Understanding the complex role of mitochondria in cancer progression and therapy response provides a wide portfolio of novel potential biomarkers and treatable targets whose therapeutic potential have to be explored for the benefit of patients, especially those developing treatment resistance.

## 2. Mitochondrial Genetics and Cancer

In mammals, mtDNA contains approximately 16.5 kb encoding 37 genes (2 rRNAs, 22 tRNAs and 13 mRNAs) (Figure 1). Thirteen proteins are essential for OXPHOS and represent only 1% of the mitochondrial proteome. The rest of the proteins important for mitochondrial functions are encoded by nDNA, translated in the cytoplasm and transferred to mitochondria [19,20,21]. Depending on energy requirements, the cell may contain only one mitochondrion but, typically, hundreds or even thousands of mitochondria are present in cells [22], with each mitochondrion containing multiple copies of mtDNA. Therefore, a certain threshold level for the mtDNA population must be compromised before the corresponding phenotype manifestation [23]. MtDNA copy number has been proposed as a promising biomarker of mitochondrial dysfunction linked to several human health disorders, including cancer, as reviewed by Cassim et al. [24].

### 2.1. mtDNA Mutations

MtDNA mutations and consequent altered mitochondrial function have been recognized as hallmarks of many cancers [25]. A direct correlation between the number of mtDNA mutations and mtDNA repair capacity has been found. Due to aberrant mtDNA repair, many cancers contain high numbers of mtDNA mutations that are not found in healthy tissues [26]. These mutations may drive processes of cell transformation and tumorigenesis, promote tumour cell proliferation and adaptation to new environments and contribute to tumour aggressiveness [27]. Therefore, mtDNA mutations represent powerful diagnostic molecular biomarkers for early detection of cancer, with the prognostic potential for detecting risk of relapse and overall patient outcome.

Cancer-related mtDNA mutations encompass point mutations, insertions, deletions and duplications. They can be found in both non-coding and coding regions and cause polypeptide mutations, mutations in rRNA and tRNA leading to defects in protein synthesis, rearrangement mutations and mutations within the control regions impacting mtDNA replication and transcription [28]. The D-loop region in mitochondrial genome represents a mutational “hot spot”, affecting mtDNA copy number and expression of mitochondrial genes.

The origins of mtDNA mutations fall into two categories: germ line mutations that predispose to cancer due to their oncogenic potential (a few mtDNA polymorphisms have been associated with increased risk of certain cancers) [29,30,31] and the tissue-specific somatic mutations that mostly participate in tumour progression [32,33]. These mtDNA mutations can be induced by both endogenous and exogenous DNA damaging agents, although it appears that the mtDNA mutational process is preferentially endogenous and linked to mtDNA replication [34]. Once the cell incorporates a somatic mtDNA mutation, it acquires a state of heteroplasmy. In heteroplasmy, both the wild-type and mutated mtDNA co-exist in the cell. Representation of mutated mtDNA is minor first but, if the mtDNA mutation provides the cancer cell with a selective advantage, it is likely to increase its proportion over time, drifting toward dominance and a homoplasmic mutant state, as observed in various diseases [35]. Chemotherapy regimens with DNA-damaging potency may also induce mtDNA mutations [36], resulting in increased mitochondrial ROS levels that interfere with chemotherapy response. An increase in ROS due to mtDNA mutations may represent a basis for one of the novel therapeutic strategy designs preferentially targeting ROS-stressed cancer cells [37,38]

At the biochemical level, cancer mtDNA mutations lead to a wide range of mitochondrial dysfunctions in terms of increased gluconeogenesis [39], a high level of glycolysis [40,41], reduced pyruvate oxidation, increased lactic acid production [42,43,44,45,46], reduced fatty acid oxidation [47,48], increased glycerol production [49,50], modified amino acid metabolism [51,52,53] and increased pentose phosphate pathway activity [54,55,56].

MtDNA mutations and/or loss of mtDNA copy number may have a causative effect in carcinogenesis via mitochondria-to-nucleus signalling [57,58], affecting diverse physiological and pathological functions; in particular, mitochondrial biogenesis, metabolism, apoptosis, cell adhesion, metastasis, mitophagy and DNA repair [59,60,61,62]. In addition to mtDNA mutations, mutations in nuclear genes encoding mitochondrial enzymes also certainly contribute to cancer development and progression [63,64,65,66,67].

### 2.2. mtDNA Repair Mechanisms

Because many anti-cancer therapies manifest their cytotoxic effects by damaging both nDNA and mtDNA, dysregulation of the DDR and repair pathways plays a critical role in treatment response. All mtDNA repair proteins are encoded by nuclear genes, synthesized in the cytosol and imported into mitochondria. Although mitochondrial and nuclear DNA repair pathways share some common proteins, they exert different organelle-specific functions, which potentially provide another new possibility for mitochondria-based therapeutic targeting. For a long time, mitochondria were believed to lack DNA repair because damaged mtDNAs were thought to be degraded [68,69]. However, despite still being underexplored, human mitochondria seem to possess almost all DNA repair mechanisms present in the nucleus, namely direct DNA damage reversal (DR), mismatch repair (MMR), base excision repair (BER), HR, non-homologous end joining (NHEJ) and microhomology-mediated end joining (MMEJ) (Figure 2). At least one novel DNA repair pathway, combining MMR and BER, is likely to exist in mitochondria. On the other hand, mitochondria lack a typical NER [70,71], opening a window for NER substrates in mitochondria-targeted therapy.

DR is the simplest way to eliminate DNA/RNA modifications without using the canonical DNA repair steps [72]. Three different DR mechanisms have so far been discovered: photoreactivation by photolyase, oxidative demethylation by ALKBH and demethylation by *O*^6^-methylguanine-DNA methyltransferase. Mitochondrial DNA repair by photolyase has been shown in yeast [73]. Although there are structural homologues of photolyase in humans, these enzymes do not show any detectable photolyase activity [74]. Mitochondrial localization of the two other DR mechanisms has not yet been reported or is inconclusive in humans [75,76,77,78].

MMR represents a DNA repair mechanism, the presence of which in mitochondria has also been documented in yeast [79], but largely independently of the known nuclear MMR factors. Human mitochondrial MMR seems to be dependent on the Y box binding protein (YB-1), which has a mismatch recognizing function [80]. Its downregulation decreased MMR activity by more than three times in mitochondrial extracts [81]. Despite localization of some MMR proteins to mitochondria, such as MutL homolog 1 (MLH1), MutS homolog 5 (MSH5) and DNA ligase 3 (LIG3) [82,83], the function of mitochondrial MMR remains an open question. In addition to MMR, YB-1 has been suggested to be involved in mitochondrial BER [71]. Due to the limited repertoire of DNA repair proteins in mitochondria compared to the nucleus, multifunctional proteins such as YB-1 are likely to be co-opted to act in multiple, and perhaps novel, mtDNA repair pathways [83]. One such potentially novel pathway combines MMR and BER in order to overcome the problem with removal of mismatched bases caused by the lack of strand bias in mitochondrial MMR [79]. Therefore, it is now almost unreservedly accepted that BER significantly contributes to repair of mismatches in mtDNA [71].

BER operates as the main DNA repair mechanism in mitochondria. This is not surprising, because mtDNA is highly vulnerable to oxidative damage from ROS due to its proximity to the electron transport chain (ETC) and due to a lack of protective histones [84]. Oxidative damage to mtDNA can occur in various forms [85] that mechanically and biochemically divide BER into two subpathways, short- and long-patch BER, which differ in the factors they engage [71,86,87,88]. BER factors, such as aprataxin (APTX), polynucleotide kinase 3′-phosphatase (PNKP) and poly(ADP-ribose) polymerase-1 (PARP-1) have also been localized in mitochondria. While both APTX [89] and PNKP [90] have clearly defined functions in mtDNA repair, the role of PARP-1 in mtDNA maintenance and repair, despite well-documented ADP (adenosine diphosphate) ribosylation of mitochondrial proteins [91,92], is still a matter of debate. Since there is no sign of the presence of X-ray, repair cross complementing 1 (XRCC1) in mitochondria [93], PARP-1 may function as a surrogate for this scaffold factor in mitochondrial BER [94]. Furthermore, PARP-1 operates in mtDNA damage signalling [94] and affects mtDNA repair through epigenetic regulation of nuclear genes encoding both the mtDNA BER and transcription factors. Finally, PARP-1 indirectly affects the integrity of mtDNA through regulation of expression of mitochondria-encoded respiratory complex subunits, COX-1, COX-2 and ND-2 [95]. Hence, PARP-1 appears to be the key factor in maintaining mitochondrial homeostasis, implying the possibility for the use of its inhibitors in efficient mitochondria targeting. In clinical oncology, apart from selective targeting of HR-deficient tumour cells, PARP-1 inhibitors represent a promising tool to modulate the size of the mitochondrial population, thereby eliminating the enormous plasticity this organelle provides for tumour cells.

As mentioned above, mtDNA is highly exposed to oxidative base damage, which may result in double-strand breaks (DSBs). In mitochondria, canonical NHEJ is not the main DSB repair mechanism [96], and it was even considered to be lacking in this organelle [97]. However, the presence of KU80-mediated DNA end-binding activity in the mitochondrial protein extract [98,99] and its ability to perform DNA end-joining in vitro [100,101] imply that mitochondria possess efficient NHEJ. MMEJ predominates over NHEJ in mitochondria; therefore, robust MMEJ repair in mitochondria might explain the frequent smaller mtDNA deletions reported in many human mitochondrial disorders, including cancer [97]. Deletions also arise due to HR [102], which, in contrast to NHEJ/MMEJ, is rare in mitochondria [75,103,104,105]. The primary function of HR in this organelle is to initiate or restart mtDNA replication [106]. However, due to fact that mitochondria lack some crucial HR promoting activities, such as RAD52-like strand annealing or resolvase activity, the existence of canonical HR in mitochondria is still rather a matter of debate.

The NER pathway seems to be totally absent in human mitochondria even though some of its individual components—e.g., the RAD23A protein, Cockayne syndrome A and B proteins (CSA and CSB, respectively) and Xeroderma pigmentosum complementing group A (XPA) protein—were detected in this organelle [71,107]. As a result, human mitochondria are defective in the removal of cyclobutane pyrimidine dimers and pyrimidine (6-4) pyrimidine photoproducts [108,109,110] and display very poor repair of intrastrand crosslinks (IaCLs), even though they efficiently remove interstrand crosslinks [76,111]. Hence, mitochondrial IaCLs may have a significant impact in a clinical context. Indeed, the recent finding regarding the strong affinity of PARP-1 for binding to IaCLs [112] hints at the possibility for alternative processing of NER substrates in mtDNA. Targeting a PARP-1-mediated, alternate NER pathway may modulate cancer treatment outcomes within modalities of DNA crosslinking agents.

DNA repair systems in mitochondria are not as poor as originally thought. However, they are much less well understood than nDNA repair systems. Deeper mechanistic details may potentially provide a platform for development of novel mitochondria-based therapeutic strategies.

## 3. Epigenetic Regulation of Mitochondrial Genes in Cancer

During tumorigenesis, epigenetic mechanisms modulate the expression of numerous transcription factors and inhibitors, contributing to regulation of cell differentiation, apoptosis or DDR. Epigenetic changes are not heritable and result in altered gene expression without any modification to the original DNA sequence. These changes encompass DNA and RNA modifications by methylation and long and short noncoding RNAs, while histone and other proteins can be post-translationally modified via processes such as acetylation, phosphorylation, methylation, SUMOylation, ubiquitination or poly(ADP-ribosyl)ation (PARylation) [113] (Figure 3). A number of factors, including environmental ones, can affect epigenetic modifications, leading to cell- and tissue-specific alterations in cellular processes. Such modifications explain the differences associated with the pathophysiology of multiple diseases, including cancer. Although multiple defects in the mitochondrial genome have been reported in cancers, the mitochondrial epigenetic landscape has often been disregarded and still remains largely unexplored [114].

Mitochondrial epigenetics, so called “mitoepigenetics”, represents a bi-directional phenomenon involving coordinated communication between the mitochondrial and the nuclear genomes [115,116], mostly via interplay between the mitochondrial-derived substrates and the nuclear epigenetics [117,118]. Variations in nuclear gene expression, as a consequence of gene mutations and epigenetic modifications, affect mitochondrial functionality via so-called anterograde regulation that includes transcription factors, co-factors or nuclear receptors [119,120]. Vice versa, mitochondrial dysfunction signals to the nucleus and affects and controls the epigenetics of nuclear genes via so-called retrograde regulation [119,120,121].

### 3.1. DNA Methylation

Mitoepigenetic modifications, such as mtDNA/mtRNA methylation and mitochondrial nucleoid modifications, have been shown to contribute essentially to tumour development. However, epigenetic modifications of mitochondria mostly relate to changes in the expression level of numerous nuclear-encoded mitochondrial genes due to nuclear DNA methylation. These genes encode proteins and enzymes required for mtDNA replication, repair, transcription and the mitochondrial respiratory chain [122]. For example, in mouse cells, Lee et al. [123] described abnormally increased DNA methylation of the nuclear-encoded mitochondrial gene for catalytic subunit A of DNA polymerase γ (POLGA), important for mtDNA replication. Increased DNA methylation of *POLGA* results in lower mtDNA copy numbers and is associated with aberrant mitochondrial functionality. mtDNA copy numbers are strictly regulated during differentiation, reflecting the ATP requirements of the cell. Cells with a high requirement for ATP generated through OXPHOS have high mtDNA copy numbers, whereas those with a low requirement have only a few copies [123]. The maintenance of low mtDNA copy number promotes aerobic glycolysis to generate ATP [124,125], leading to cell proliferation and prevention of differentiation [126,127]. Accordingly, highly proliferative cancers and pluripotent cells possess low mtDNA copy numbers and their DNA remains extensively methylated [128]. mtDNA copy number is further regulated by balanced cycles of mitochondrial fusion/fission, ensuring appropriate distribution throughout the cell [19,129]. Defects in fusion were positively associated with mtDNA depletion in several studies. Heart-specific knockout of mitofusins (Mfns) in mice and Mfns knockout mouse embryonic fibroblasts (MEFs) showed reduced levels of mtDNA copy number [130]. Similarly, disruption of the *MFN1* and *MFN2* genes in mouse skeletal tissue led to mtDNA loss and severe mitochondrial dysfunction [131]. In another report, reduction of mtDNA copy number was induced by optic athrophy protein 1 (*OPA1)* mutations [132]. Interestingly, inhibition of fission by dynamin related protein 1 (*DRP1)* knockout or silencing did not influence mtDNA copy number [133,134,135].

Mitochondria were originally considered to lack their own DNA methylation machinery [136]. Using different methodological approaches, mtDNA methylation has been evaluated and predicted to occur at much lower levels than in the nucleus [137], with an approximate range from 1 to 20% [138]. Employing bisulfite sequencing [139,140] and mtDNA methylation determination and evaluation, the secondary structures of mtDNA and its circular structure were claimed as possible causes of overestimation of methylation signals [138]. Increasing evidence suggests that changes in DNA methylation do occur in mtDNA, implying a valid role for methylation in mitochondrial genome replication and gene expression control [141,142,143]. On the other hand, stimulating mtDNA replication results in increased methylation [144], which suggests that mtDNA methylation can also be a feedback regulatory mechanism that maintains mtDNA copy number [143].

Cancer development and progression often involve a low level of the total nDNA methylation but encompass hypermethylation of the tumour suppressor gene promoters and hypomethylation of the oncogene promoters [145]. When DNA is methylated, cytosine is converted to 5-methylcytosine (5mC) by DNA methyltransferase (DNMT) (reviewed in [146]). Cytosine can be methylated either on the fifth carbon of the pyrimidine ring (C5) or at the exocyclic NH_2_ group at position 4 (N4). Similarly, adenine methylation (6mA) can occur on the exocyclic NH_2_ group at position 6 of the purine ring (N6). As one of its functions, the restriction-modification systems of prokaryotes use such DNA modification as a discriminatory tag preventing the restriction of self-DNA and allowing targeting and discarding of unmodified non-self DNA [147,148]. 6mA has also been observed in higher eukaryotes, including humans [149,150].

In the nucleus, cytosine methylation predominantly occurs in CpG dinucleotides located within CpG islands. Given that CpG islands are virtually absent in mitochondria, mtDNA methylation occurs only on CpG dinucleotides [28]. Jang et al. [151] also observed methylation of nDNA within non-CpG sites; namely, the CpA, CpT and CpC positions in specific cell types, such as stem cells, oocytes, neurons or glial cells. All these modifications can affect promoters, enhancers, transcription factors and transcriptional elongation and gene splicing, thereby impacting gene expression regulation [152,153]. DNA methylation at the transcriptional start sites alters chromatin structure to inhibit transcription factor binding and/or recruitment of transcription repressors to methylated nucleotides to repress gene expression [154].

In mtDNA, non-CpG methylation and adenine methylation have also been observed [155,156,157]. Sharma et al. [28] reported a >8000 times higher level of 6mA in the human mitochondrial genome than in the nuclear genome and described its uniform distribution throughout mtDNA. The increase in 6mA levels in mtDNA compared to nDNA indicates that mtDNA methylation might occur predominantly on adenine, which supports the endosymbiotic theory of mitochondrial origin [157].

Demethylation, a reverse process leading to removal of methylation marks from DNA or histones, is performed by Ten-Eleven Translocation (TET) methylcytosine dioxygenases or Jumonji catalytic (JmjC) domain-containing demethylases [158]. To remove 5mC, TET dioxygenases convert it to 5-hydroxymethylcytosine, which is followed by oxidation to 5-formylcytosine and 5-carboxylcytosine prior to removal by BER [159].

Mitochondrial expression of both DNMTs and demethylases in different cell types is controversial and appears to be cell- and tissue-specific [146,160]. Currently, DNMT1, DNMT3a, DNMT3b and TET1 and TET2 enzymes have all been identified within the mitochondria [122,146,155,161]. Although the methyltransferase responsible for adenine methylation in mitochondria has not been identified, the organelle contains the demethylase Fe(II)/α-ketoglutarate-dependent dioxygenase 1 (ALKBH1), which plays a role in OXPHOS regulation [157].

In solid tumours, the absence or very low levels of mtDNA methylation have previously been reported [143,162,163]. However, recent work employing novel, high-throughput next-generation sequencing methods and genome-wide analysis of mtDNA methylation across different cell lines and tissue samples refutes these observations and points to non-random, reproducible patterns of mtDNA methylation, but predominantly in a non-CpG context [164]. In addition, different mtDNA methylation profiles have been observed in cancer cells derived from liver and breast cancer when compared to their normal tissue counterparts [164,165]. However, van der Wijst et al. (2017) provided solid evidence for decreased mtDNA gene transcription and an abundance of mitochondria-encoded transcripts due to increased GpC methylation. Others proposed an alternative indirect modulation of the gene expression via modulation of mtDNA replication [155,156,166] due to fact that the D-loop provides an open DNA structure for binding the proteins involved in mtDNA replication or transcription [167].

Several studies point to higher methylation rates in the D-loop region and more frequent hypermethylation events in cancer compared to non-cancerous samples, as well as in metastatic compared to primary tumour samples. This confers a selective advantage for tumour growth and metastatic potential [156,168,169,170,171,172]. A study by Liu et al. (2022) on kidney clear cell carcinoma (RCC) bone metastasis provided evidence of a higher methylation rate at the D-loop region of mtDNA in bone metastases than in primary RCC tumour cells. These authors observed a significant increase in gene expression for many mitochondrial genes after demethylation with the DNMT inhibitor 5-azacitidine. Consequently, reduction of mtDNA methylation resulted in decreased ROS production and intracellular ATP levels, supressing bone metastatic tumour progress in RCC. This study provides evidence of a direct association between increased mtDNA methylation and bone metastatic tumour growth in RCC.

Different structural variants and altered gene expression of mtDNA have been reported in 38 common cancer types [172]. In a hypoxic tumour environment, activation of hypoxia inducible factor 1α (HIF-1α) together with epigenetic reprogramming were described as the main contributors. It was shown that hypoxia not only activates HIF-1α but also triggers approximately 50% of all hypermethylation events in solid tumours [168]. Nevertheless, direct regulation of mtDNA gene expression by mtDNA methylation remains an open question [169].

### 3.2. RNA Methylation

Important epigenetic modulations also occur via posttranscriptional modifications of multiple mitochondrial RNAs (mtRNAs: mt-rRNAs, mt-tRNAs, mt-mRNAs and multiple non-coding RNAs) present in the mitochondrial matrix [173]. These mtRNAs are processed from a single polycistronic RNA precursor transcribed from mtDNA. Posttranscriptional modifications of mtRNAs are very important for processing, maturation, ribosome biogenesis and assembly [174], as well as for translation initiation within mitochondrial RNA granules [175,176,177], which are dynamic structures juxtaposing nucleoids assembled around the newly synthesized primary transcripts. Once they become detached, granules locate within the inner mitochondrial matrix and trigger events of mitochondrial gene expression [178].

Different modifications of mtRNA transcripts and their accumulation in tumour tissues were associated with tumorigenesis and drug resistance [179,180]. The “mitoepitranscriptome” represents a new concept describing the dynamic regulation of gene expression by modified mtRNAs. In several tumour tissues, altered methylation levels of mt-tRNAs have been described to affect maturation and mitochondria-mediated metabolism [181]. In normal tissues, RNA processing pathways are specifically related to mt-tRNA methylation levels; however, these connections are lost in tumours. Large differences in mt-tRNAs methylation can predict a poorer prognosis, as shown in cohorts of patients with breast invasive carcinoma, colon adenocarcinoma, head and neck squamous cell carcinoma, RCC, kidney renal papillary cell carcinoma, liver hepatocellular carcinoma and lung, prostate and stomach adenocarcinoma [182].

### 3.3. Histone Modification

In the nucleus, histones are basic proteins that are targets for post-translational modifications, such as acetylation and methylation. Acetylation is controlled by the balanced activity of histone acetyltransferase (HAT), which adds an acetyl group at the N-terminus of lysine, resulting in chromatin unfolding and gene transcription [183], and by histone deacetylases (HDACs), which play the opposite role. In tumours, besides competition for acetylases and deacetylases in epigenetic regulation of the gene transcription, histone acetylation and de novo synthesis of fatty acids compete for available acetyl-CoA pools. Carboxylation of acetyl-CoA, the rate-limiting reaction in de novo synthesis of fatty acids, is catalysed by the acetyl-CoA carboxylase (ACC1) [184]. Attenuation of ACC1 activity increases acetylation of chromatin histones and alters transcription via regulation of availability of acetyl-CoA for HATs [185]. Another modification of histones that contributes to regulation of transcriptional activity is represented by methylation, an addition of the methyl group to the N-terminal lysine [186].

In mitochondria, mtDNA is found in packed mtDNA nucleoids, protein-DNA structures similar to histones in nuclear DNA nucleosomes. The protein scaffold of mtDNA nucleoids can be epigenetically and post-translationally modified, just like histones in the nDNA nucleosomes. There are more than 50 nucleoid-related proteins, but mitochondrial transcription factor A (TFAM) is the only nucleoid-associated protein that functions as a histone-like factor. Post-translational modifications of TFAM have emerged as essential for mtDNA replication and transcription [187], and its modifications by acetylation, phosphorylation and ubiquitination affect mtDNA packaging [167,188,189,190]. The level of *TFAM* expression was shown to correlate positively with the progression of multiple cancers; e.g., colon cancer [191], melanoma [192], hepatocellular carcinoma [193], breast cancer [194] and non-small cell lung cancer (NSCLC) [195]. Sun et al. [196] suggested that tumour cells produce more TFAM to achieve sufficient compaction of the higher mtDNA copy numbers found in tumours compared to normal tissues. It is possible that mtDNA in cancer cells may be tightly wrapped by the increased amount of TFAM, which decreases expression of mtDNA-encoded genes related to the electron-transport chain and OXPHOS and promotes tumour cells’ use of aerobic glycolysis [196].

### 3.4. Epigenetic Role of Mitochondrial Metabolites

Besides production of up to 90% of cellular ATP, mitochondria function as a source of metabolic activity, producing numerous metabolites including ROS, fatty acids, one-carbon metabolic intermediates, TCA cycle metabolites and ketone bodies, which can epigenetically modify cellular phenotype. Chen et al. [197] identified 346 distinct metabolites predicted to be present in human mitochondria and measured the matrix concentrations of more than 100 of those that potentially regulate the enzymes modifying nDNA and histones to modulate gene expression. These metabolites can control gene expression by fuelling the addition of chemical groups or epigenetic marks to DNA or DNA-associated proteins. Such marks influence transcription factor or repressor interactions within chromatin, consequently enhancing or repressing gene expression. Evidence establishing the regulatory effect of mitochondrial metabolites on cellular metabolism, tissue repair and disease progression has been provided by incorporating mitochondria into recipient cells [198].

Histone acetylation is dependent upon the availability of acetyl-CoA, which accumulates within the nucleus, cytoplasm and mitochondria [199]. In cancer cells, a plethora of metabolic pathways provide the acetyl-CoA pool, and the metabolic source of acetyl-CoA seems to dictate which transcriptional program will be favoured. Besides glucose, the major source of acetyl-CoA, metabolic fuels, such as glutamine, fatty acids, acetate or lactate and nuclear glycogen, represent alternate sources of acetyl-CoA affecting histone acetylation [200,201]. Although acetyl-CoA has traditionally been regarded as obligatorily mitochondrial in origin, a non–mitochondrial source of acetyl-CoA (from outside the cell) has been proposed. Due to the activity of the ATP-citrate lyase (ACLY) [202], acetyl-CoA can be generated from citrate (derived from glutamine) [203] or from the activity of acetyl-CoA synthetase (ACSS). In this case, a specific isoform of ACSS, the acetyl-CoA synthetase short chain family member 2 (ACSS2), active under metabolic stress, produces acetyl-CoA from acetate [204]. The nuclear glycogen-derived pyruvate can be metabolized to acetyl-CoA by the pyruvate dehydrogenase complex directly in the nucleus.

Some tumours, such as glioblastomas [205] or hepatocellular carcinomas [206], upregulate the expression of *ACSS2* to capture acetate. During hypoxia, hepatocellular carcinoma cells increase *ACSS2* expression by nearly fivefold [207], enhancing the rate of acetate utilization and using the resultant acetyl-CoA to increase fatty acid synthesis and acetylation of histone H3 at K9, K27 and K56 [204], which affects transcription. An increase of highly phosphorylated ACSS2 (at Ser659), associated with its elevated nuclear translocation, was observed in NSCLC cells [208,209]. However, increased rates of metastasis and poor prognosis in some cancer patients correlated with downregulation of *ACSS2* expression [210].

Conversely, acetyl marks are removed by Zn^2+^-dependent (HDACs 1-11) or NAD^+^- requiring (sirtuins, SIRT1-7) HDACs [211,212]. Inhibition of HDACs by different mitochondrial metabolites can result in histone H3/H4 tail hyperacetylation [213], transcriptional changes and activation of the oxidative stress transcription factors forkhead box O3 (FOXO3A) and metallothionein-2 (MT2) [214,215], inducing tumorigenesis [216]. In hepatocellular carcinoma cells, increased H3K9 acetylation was observed after depletion of SIRT1 and hyperacetylation, and reduced trimethylation at H3K9 under NAD^+^ depletion reduced cell proliferation in human cancer models [217].

Histone-tail amino acids, mostly arginine and lysine, are also subjected to methylation [218,219]. The activity of DNMTs depends upon mitochondrial function. These enzymes require S-adenosyl methionine (SAM) synthesized from ATP, one-carbon cycle intermediates and methionine [220]. On the other hand, several TCA cycle metabolites regulate gene expression via competitive inhibition of demethylase activity. In cells with mutations in fumarate hydratase and succinate dehydrogenase, an accumulation of fumarate and succinate inhibits TET demethylases and 5-hydroxymethylcytosine formation, resulting in reduced histone demethylation, hypoxia-inducible pathway activation [221] and tumorigenesis [222]. Elevated levels of tricarboxylic acid (TCA) cycle-associated metabolite, 2-hydroxyglutarate (2HG), can promote tumorigenesis [223], enhance mammalian target of rapamycin (mTOR) signalling [224] and alter T-cell differentiation [225]. The cellular concentrations of TCA cycle metabolites, alpha-ketoglutarate (αKG) and succinate, are dependent upon mitochondrial activity and are important regulators of DNA and histone methylation and cell fate. The αKG-to-succinate ratio can cause phenotype switching in cells. In macrophages, a high αKG-to-succinate ratio causes demethylation of H3K27me3 marks and drives M2 macrophage differentiation for anti-inflammatory and tissue repair responses, whereas a low ratio promotes lipopolysaccharide-sensing M1 macrophages and a pro-inflammatory response [226]. The elevated ratio of αKG-to-succinate can maintain DNA/histone demethylase function and pluripotency, while reduction of this ratio impairs TET demethylase activity, increases trimethylation and decreases monomethylation of H3K9, H3K27, H3K36 and H4K20 and promotes cell differentiation [227].

These examples show that specific mitochondrial metabolites can differentially regulate epigenome-modifying enzymes and consequent cellular processes, reflecting the cell state or responding to microenvironmental cues.

### 3.5. Methylation of Mitochondria-Related Nuclear Genes

The expressions of the multiple nuclear encoded genes involved in mitochondrial quality control and cellular response to hypoxia are under different epigenetic controls. For example, in multiple myeloma, *MFN1* and *MFN2* were hypermethylated, while no significant change in methylation levels was observed in the mitochondrial fission (*FIS1)* gene [219]. In breast cancer, decreased expression of *MFN2* was observed, accompanied by the hypermethylation of its promoter. Subsequent demethylation with 5-aza-2′-deoxycytidine restored normal *MFN2* expression and suppressed cancer growth and metastasis [220]. Aberrant promoter methylation of the Parkin (*PRKN)* gene is another common feature of breast cancer cells. It has been shown that Parkin promoter methylation strongly downregulates Parkin protein and patients with Parkin methylation or Parkin loss suffer worse outcomes [221]. Inactivation of Parkin due to its promoter hypermethylation was detected in nasopharyngeal carcinoma [222]. In glioblastoma cells, HIF-1α was indicated to be methylated in vivo by methyltransferase G9a, inhibiting its transcriptional activity and downstream target gene expression [223]. Tumour-associated CpG demethylation might represent a mechanism of epigenetic autoregulation of HIF-1α expression. Koslowski et al. [224] found that the HIF-1α promoter itself possesses a hypoxia response element (HRE), which is normally repressed by methylation of a CpG dinucleotide in normal cells. In contrast, the HRE is frequently and aberrantly activated in colon cancer cell lines and primary colon cancer specimens, where demethylation enables HIF-1α binding to its own promoter, resulting in autotransactivation of the HIF-1α expression.

In cancer development and progression, all of the abovementioned epigenetic modifications relate to the maintenance of genome stability and apoptosis. Exploration of mitochondrial contributions to epigenetic regulation of nuclear-encoded genes may have been underestimated in the past, but today it could help to elucidate regulatory events related to the tumour heterogeneity of the histological subtypes and provide clues regarding therapy resistance and development of new targeted therapies.

## 4. MiRNA Regulation of Mitochondrial Genes

MicroRNAs (miRNAs or miRs) represent a class of small (~22 nucleotides) non-coding RNAs transcribed from DNA sequences. They posttranscriptionally regulate gene expression and thus affect various cellular pathways, including those in mitochondria [228,229]. Modulation of mitochondrial processes by miRNA binding to mRNA occurs in the cytoplasm, or miRNAs are translocated into mitochondria, where they can directly target mitochondrial mRNAs. Moreover, human mitochondria themselves have been shown to possess miRNA sequences in their DNA, although their transcription from the mitochondrial genome remains to be fully clarified. MiRNA localizing in the mitochondria are referred to as “mitomiRs” and their activity mostly involves regulation of OXPHOS and ROS generation [229,230,231].

In cancer, mitomiRs were shown to directly or indirectly regulate mitochondrial processes. For example, colorectal cancer expression profiles of mitogenome-associated miRNAs showed increased levels of miR-24, miR-181, miR-210, miR-21 and miR-378 [232]. The oncogenic effects of miR-210, miR-155, miR-21, miR-224 and miR-373, and the anti-tumour action of miR-29a, miR-128, miR-342, miR-30a, miR-340, miR-18a and miR-224, have been associated with breast cancer pathologies (reviewed in [233]). Microarray studies found significant changes in the expression of miR-7, miR-153, miR-21, miR-34a and miR-128 in malignant gliomas [234].

Regarding metabolic reprogramming and OXPHOS regulation, miR-661 was shown to directly target cytochrome c1 and was downregulated in osteosarcoma cells [235]. Furthermore, the glycolytic phenotype of glioma cells can be disrupted by miR-128-3p overexpression, which is able to modulate pyruvate dehydrogenase kinase 1 [234]. On the other hand, pyruvate dehydrogenase kinase 2 was suggested to be a target for miR-422a when overexpressed in gastric cancer [236]. The tumorigenicity of giant cell tumours of bone was demonstrated to be significantly reduced by restoration of miR-127-3p and miR-376a-3p expression, and this positive effect seems to be partially mediated via cytochrome c oxidase assembly factor 1 homolog (COA1) and protein disulphide isomerase family A member 6 (PDIA6) [237]. A later study by the same group confirmed the role of miR-127 and miR-376a in tumour suppression in osteosarcoma cells [238]. In contrast, osteosarcoma cell proliferation can be enhanced via targeting of ubiquinol-cytochrome c reductase core protein 1 (UQCRC1) with miR-214-3p [239]. In gastric cancer, the miR-370/UQCRC2 axis positively regulates the epithelial-mesenchymal transition (EMT) signalling that affects tumour proliferation and metastasis [240]. However, there is still limited information on direct regulation of mitochondria-encoded proteins by mitomiRs in cancer. Jung et al. [241] confirmed that miR-181c, able to regulate the mitochondrial genome in rat cardiomyocytes, interacted with the mitochondrial gene encoding cytochrome c oxidase subunit 1 (MT-CO1). Reduction of miR-181c-mediated MT-CO1 was associated with mitochondrial membrane potential disruption, ATP reduction and adenosine monophosphate (AMP)-activated protein kinase-α (AMPKα) activation in human colon cancer cells [241]. Expression of mitomiR miR-26a is decreased in prostate cancer compared to normal prostate cells, but its overexpression after transfection with miR-26a mimics significantly enhanced apoptosis via MT-CO2 inhibition [242]. A recent report showed miR-181a-5p potentially targeting the *MT-CO2* and cytochrome b (*MT-CYB*) genes in human hepatocellular carcinoma. Overexpression of miR-181a-5p reduced the level of MT-CYB and MT-CO2, impaired mitochondrial function and promoted glycolysis, leading to higher proliferation and metastasis [243]. Differential expression microarray analysis identified downregulation of miR-5787 in cisplatin (CDDP)-resistant tongue squamous cell carcinoma (TSCC), resulting in suppression of MT-CO3 translation and a metabolic shift from OXPHOS to glycolysis [244]. The CDDP-resistant phenotype of TSCC cells was found to be regulated by miR-2392. In chemoresistant patients, tumour expression of miR-2392 was increased and partially regulated transcription of mitochondrial genes in a splicing-competent Argonaute 2 (AGO2)-dependent manner [139].

Several studies have indicated a connection between mitomiRs and mitochondrial dynamics. *MFN2* was shown to be a direct target of miR-195 in breast cancer cells. Mfn2 levels were significantly decreased when miR-195 was overexpressed and mitochondria became round, small and fragmented; however, miR-195-induced apoptosis was Mfn2-independent [245]. In other study, negative regulation of *MFN2* by miR-125 led to induction of mitochondrial fission and promotion of pancreatic cancer cells’ apoptosis [246]. Zhou et al. [247], investigating the role of miR-761 in hepatocellular cancer cells in the context of mitochondrial dynamics, revealed that miR-761 overexpression directly downregulated Mfn2 and increased cell migration and invasion. When a miR-761 was inhibited, cancer cells were more receptive to apoptosis by upregulation of Mfn2 [247]. In osteosarcoma cells, miR-19b was reported to directly target *MFN1* and suppress its anti-proliferative activity. On the other hand, inhibition of miR-19b promotes Mfn1-induced apoptosis. Several miRNAs have been shown to affect mitochondrial dynamics in chemoresistant cancer cells. MiR-98 and miR-148a-3p have been connected with regulation of Drp1 in resistant bladder and gastric cancers [248,249]. In TSCC, CDDP sensitivity was dependent on miR-483-5p regulation of Fis1 expression through a breast cancer type 1 susceptibility protein (BRCA1)–miR-593-5p–mitochondrial fission factor (Mff) axis [250,251].

MitomiRs, as far-reaching regulators, have been shown to also be directly linked to various mitophagy-associated proteins. Mitophagy represents a selective form of autophagy that cells use to remove dysfunctional or damaged mitochondria. In phosphatase and tensin homolog (PTEN)-induced kinase I/Parkin (PINK1/Parkin)-mediated mitophagy, miR-27a, miR-27b, miR-181a and miR-218 have been shown to regulate PINK1 and Parkin, but newer reports also indicate the involvement of miR-34a-5p, miR-103a-3p and miR-155 [252,253,254,255,256]. In cancer, miR-181a suppresses Parkin-mediated mitophagy and sensitizes neuroblastoma cells to mitochondrial uncoupler-induced apoptosis [257]. There is limited information on miRNA regulation of mitophagy receptors. MiR-137 has been reported to impair mitophagy in response to hypoxia via direct regulation of NIX (or BCL-2/adenovirus E1B 19 kDa-interacting protein 3 like (BNIP3L)) and Fun14 domain containing 1 (FUNDC1) receptors, and miR-145 has been suggested to be an upstream regulator of BNIP3-dependent mitophagy [258,259].

Several studies confirmed the ability of miRNAs to regulate proteins engaged in intrinsic apoptosis. In breast cancer, miR-195, miR-24-2 and miR-365-2 were shown to act as negative regulators of BCL-2 that enhance apoptosis [260]. Slattery et al. associated miR-203a with BCL-2-mediated apoptosis in colorectal cancer [261]. Chronic lymphocytic leukaemia exhibits deletion or downregulation of miR-15a and miR-16-1. Expression of these mitomiRs inhibits BCL-2-induced apoptosis [262]. Overexpression of miR-519d enhances the sensitivity of CDDP-resistant breast cancer stem cells and mediates apoptosis through induced myeloid leukaemia cell differentiation protein (MCL-1)-dependent mitochondrial pathway [263].

In hypoxia, HIF-modulated miR-210 is the major regulator of metabolism. MiR-210 reduces the ETC activities of complexes I and IV by disrupting electron flow through these chains and inducing the formation of ROS during hypoxia [264,265]. Overexpression of miR-210 was documented in pancreatic, head and neck, breast and lung cancers [266]. Similar to miR-210, miR-323 is upregulated in hypoxia, and its low levels are connected to longer survival in human glioblastoma patients [267]. Although miR-137 functions as a tumour suppressor [268], it also plays a role in hypoxia and inhibition of ROS accumulation; miR-137 directly downregulates the expression of the hypoxia-mediated mitophagy receptors FUNDC1 and NIX, leading to suppression of apoptosis and mitophagy [253]. Overexpression of miR-137 suppresses cell proliferation and migration [269,270]. MiR-216b inhibits proliferation and cell invasion in several types of cancer and is downregulated in hypoxia. Low levels of this miRNA were associated with poor prognosis [267].

Mir-663 is necessary for stability of the respiratory complexes. It regulates expression of nuclear proteins involved in ETC in complexes I, II, III and IV. However, along with disruption of OXPHOS, expression of miR-663 is downregulated, showing mitochondria-to-nucleus retrograde signalling involving ROS. Increased miR-663 expression in patients with breast tumours correlated with increased survival [271]. Overexpression of miR-128 reduced cell proliferation, angiogenesis and tumour growth through effects on HIF-1 and vascular endothelial growth factor (VEGF) [272].

MiR-650, miR-665, miR-640, miR-1182, miR-1203, miR-661 and miR-1204 were all shown to be important in testicular cancer. MiR-650 and miR-665 were associated with the PI3K/AKT and Wnt/β-catenin signalling pathways, which participate in growth, migration and invasion of cancer cells. MiR-650 is downregulated in cancer and associated with inhibition of cell growth and invasion. This downregulation can activate AKT pathways and promote cell migration and proliferation. MiR-665 inhibits c-MYC and thus suppresses tumorigenesis. Tumour suppressor miR-665 is downregulated in cancer. Low levels of miR-661 and miR-640 are associated with poor cancer prognosis. MiR-640 affects the VEGF receptor 2-mTOR pathway. Upregulated miR-1204 promotes proliferation, glucose uptake and ATP production and correlates with tumour size [273].

There is a significant number of miRNAs that affect mitochondrial processes [274,275,276]. These mitomiRs, both oncogenic and tumour-suppressive, have been proposed as the key regulators of cancer-related processes, and their therapeutic targeting can be suggested as one of the key emerging novel diagnostic and therapeutic tools [277].

## 5. Targeting Mitochondria to Combat Cancer

Given the massive involvement of mitochondria in cancer, therapeutic targeting of their genetic, epigenetic and miRNA landscape has a clear justification. Several molecular mechanisms of mitochondrial retrograde signalling have been revealed, such as those mediated by ROS, NAD^+^/NADH and ATP, as well as those involving mitochondrial unfolding protein response, Ca^2+^ gradient and calcineurin, AMP-activated protein kinase signalling and the “mipigenetic” process (mitochondrial–nuclear intergenomic crosstalk at the genetic and epigenetic level) [57]. The retrograde signalling represents a causal factor in tumorigenesis but also induces invasive behaviour in tumour cells and contributes to tumour progression. Not surprisingly, targeting retrograde signalling mechanisms is of a great interest, as it provides another promising strategy for the development of selective cancer therapy.

In view of the regulation of the nucleo-cytoplasmic pool of acetyl-CoA, new promising therapeutic approaches have been envisioned. In this regard, the potential of *ACLY* or *ACSS2* gene expression and activity modulation has been proposed. High levels of *ACLY* expression associated with high proliferation rates have been detected in many types of tumours [278], and this enzyme has been investigated as a prognostic factor for several cancers [279,280]. Wei et al. (2021) identified significantly upregulated ACLY in CDDP-resistant ovarian cancer cells and managed to sensitize the resistant cells by *ACLY* knockdown. ACLY has been proposed as the key enzyme regulating acquired CDDP resistance in ovarian cancer and, therefore, as a suitable novel target for sensitization of ovarian tumours to platinum agents [278,281]. Another promising target for cancer therapy is ACSS2, the enzyme responsible for capturing acetate, the major source of acetyl-CoA, especially in hypoxia. *ACSS2* is highly expressed in numerous tumours and several studies have shown that genetic depletion or pharmacological targeting of ACSS2 inhibits tumour growth in breast, prostate, liver, pancreatic, ovarian and skin cancers and glioblastoma [206,207,208,282,283].

MiRNAs modulate mitochondrial function via direct targeting of the mRNA of nuclear-encoded mitochondrial genes [284]. Via their non-canonical functions, miRNAs may act as chromatin and transcriptional regulators in the nucleus (i.e., miR-584-3p and miR-26a-1) [285] or as translational activators in mitochondria [286]. In the mitochondrial genome, some mitomiRs’ seed sequences have been detected in silico, supporting regulation of mitochondrial transcripts encoded by mitomiRs [287,288]. In principle, miRNA-targeted therapies are based on either replenishing the tumour-suppressive miRNA mimetics, which enables restoration of the lost or downregulated tumour suppressor miRNA [289], or administration of miRNA antagonists, antimiRs, targeting oncomiRs [290,291]. Accordingly, novel treatments based on targeting miRNAs have been proposed. For example, nanoparticles with miR-634 mimics caused a significant reduction in pancreatic tumour growth [292]. Engineered extracellular vesicles released from the mesenchymal stem cells carrying miR-379 have been systematically administered to inhibit tumour growth in breast cancer [293]. Similarly, marrow stromal cell exosomes carrying the miR-146b expression plasmid were successfully employed as an anti-tumour therapy in a rat model of glioblastoma [294]. Nanoparticles carrying anti-miR-21 were able to selectively target triple-negative breast cancer (TNBC) cells to reduce miR-21 expression and activate apoptosis and proliferation control [295,296]. Inhibition of miR-150 and miR-638 was found to be efficient in reducing primary melanoma growth, as well as metastases [297]. Treatment with combined oxaliplatin and miR-204-5p on silica nanoparticles significantly decreased growth of colon cancer [298]. Anti-miR-21 and miR-100 on gold–iron oxide nanoparticles loaded with PEG-T7 peptide increased the overall survival rate in mice with glioblastoma multiforme when used in combination with systemic temozolomide administration [299]. Today, there are numerous clinical trials investigating anti-miRNA sequences as miRNA-based cancer therapy, as monotherapy or as a combinatorial therapy. Targeting mitomiRs has huge therapeutic potential and, in parallel, mitomiRs represent a novel group of suitable diagnostic and prognostic markers.

With regard to the therapeutic focus on mtDNA mutations and their repair, it is evident that one of the promising, effective ways to prevent the onset and progression of human cancers associated with this organelle is elimination of the mtDNA mutations. As none of the DNA repair factors are encoded in mtDNA, efficiency in mtDNA repair can only be achieved by potential direction of more nuclear DNA repair factors to the mitochondria, although a few DNA repair factors possess mitochondrial targeting sequences. Indeed, targeting of DNA repair to mitochondria was shown to not only enhance the repair of mtDNA lesions but also increased the viability of treated cells and protected them against induction of apoptosis [300,301]. However, once an mtDNA mutation is fixed, its negative effect can be corrected only by introducing a construct carrying and expressing the wild-type version of the affected gene. Hence, one promising therapeutic approach for patients with mtDNA mutations is based on expression of the relevant proteins that are fused at the N-terminus with the mitochondrial targeting sequence. Unfortunately, no successful method has yet been introduced for clinically complementing mitochondrial dysfunction caused by mtDNA mutation in human mitochondrial disorders.

## 6. Therapeutic Potential

Over the last few years, mitochondria have proven to be an intriguing target for anti-cancer drugs. Indeed, due to their great clinical potential, anti-cancer agents targeting these organelles have become a highly prioritized focus of current cancer research. The exceptional potential of mitochondria to act as an anti-cancer target is strengthened by the facts that the abovementioned tumour-specific somatic mutations in mtDNA vary across tumour types [302] and, importantly, that tumours of the same type but from different individuals may vary remarkably in the mutations functionally affecting mitochondria [303]. Both facts point to an extraordinary possibility for the personalisation of anti-cancer therapy via targeting of mitochondria. Therefore, anti-cancer agents acting via mitochondrial destabilization, collectively referred to as mitocans (an acronym for “mitochondria and cancer”), represent one of the most innovative therapeutic approaches to drug targeting for the “next generation”. These agents have been divided into several classes based on the mode and site (from the surface of the outer mitochondrial membrane to the mitochondrial matrix) of action. Individual classes consist of the hexokinase inhibitors, compounds targeting the BCL-2 family proteins, thiol redox inhibitors, drugs targeting the voltage-dependent anionic channel and adenine nucleotide translocase, electron redox chain-targeting drugs, drugs targeting the TCA cycle, drugs targeting mtDNA and delocalized lipophilic cations (DLCs) targeting inner mitochondrial membrane [303]. Selected examples of mitocans and their biological effects are described below, with particular focus on those covalently linking DLCs with widely used chemotherapeutics.

The highly negative plasma and mitochondrial membrane potentials (MMPs) (30–40 and 120–180 mV, respectively) permit 5–10 times higher concentrations of cations to be present in mitochondria than in the cytosol and 100–1000 times higher concentrations in the mitochondrial matrix than in the cytosol [304,305]. Based on this and on the fact that cancer cells have a more hyperpolarized MMP (app. 220 mV) [306], several delocalized lipophilic cations (DLCs) have been introduced into experimental cancer research to improve mitochondrial uptake of anti-cancer drugs of interest [304]. At the beginning, however, the use of DLCs in mitochondrial targeting was primarily developed to study mitochondrial physiology and dysfunction and the interaction between mitochondria and other subcellular organelles. Subsequently, many other applications for DLC-mediated mitochondrial targeting were revealed including those aimed at developing new therapy strategies in the field of cancer [304,307].

Triphenylphosphonium cation (TPP^+^) is the most extensively used mitochondria targeting DLC structure and can easily pass through phospholipid bilayers because its charge is dispersed over a large surface area and the potential gradient drives its accumulation into mitochondrial matrix. When administered orally, it is able to pass from the gut to the bloodstream, within which it rapidly redistributes into organs. Direct intravenous injection can also be used to deliver the TPP^+^-conjugated compounds to the mitochondria within cells in an organism. Importantly, orally administered TPP^+^-conjugated compounds can be bioavailable, as shown for TPP^+^ coupled to a coenzyme Q or vitamin E derivative in mice, where significant doses of these compounds could be fed safely over long periods, leading to steady-state distributions within several tissue/organ types. This indicates that therapeutic concentrations of the TPP^+^-conjugated compounds can be reached by oral administration in tissues affected by mitochondrial dysfunction [308], with the caveat that the toxic effects due to non-specific disruption to mitochondria caused by accumulation of large amounts of the DLCs may represent the major factor limiting the amount of the compounds that can be administered safely. Tethering TPP^+^ to metformin significantly reduces human pancreatic carcinoma cell proliferation and this effect is achieved through reduced oxygen consumption accompanied by ROS formation [309]. In addition, administration of a TPP^+^ derivative of chlorambucil (Mito-Chlor) causes an 80-fold increase in cell killing in breast and pancreatic cancer cell lines and delays tumour progression in a mouse xenograft model of human pancreatic cancer [310]. TPP^+^ was also conjugated to doxorubicin (DOX) to generate TPP-DOX derivative. Enhanced cytotoxicity and apoptosis were observed for TPP-DOX compared to free doxorubicin in the MDA-MB-435 cancer cell line. These effects were more pronounced in doxorubicin-resistant than in doxorubicin-sensitive cells, suggesting that preferential distribution of doxorubicin to the mitochondria can revert drug resistance in tumour cells [311].

In addition to TPP^+^, a couple of other DLCs (either alone or attached to the spacer) have already been used to target various anti-cancer pharmacophores into mitochondria in order to destabilize them in cancer cells, with F16, rhodamine (B and 101), rhodacyanine MKT-077, dequalinium, heterocyclic aromatic cations, natural and synthetic mitochondria-targeting peptides and mitochondria-targeted nanoparticle (NP) vesicles being the most important ones. Both natural and synthetic compounds have functionally been conjugated to these DLCs. As a consequence, the resulting biological effects of the conjugates usually significantly exceed those of their precursors. As an example, F16-conjugated natural pentacyclic triterpenoids showed a considerable enhancement of antitumour action in comparison with the parent compounds and a markedly higher cytotoxic effect against tumour cell lines over healthy fibroblast cells [312]. The same findings were reported for triterpenoids linked to TPP^+^ [313,314,315,316,317]. Pentacyclic triterpenoid-conjugates with rhodamine also displayed increased cytotoxic effects in cancer cells compared to non-malignant fibroblasts [318,319].

Although detailed information on how DLC conjugates mediate mitochondria-targeted cytotoxic effects that are highly cancer-specific is still missing, a couple of mechanisms have been suggested. Depending on the particular DLC compound and its conjugate, these include a surface-active effect on mitochondrial membranes causing organelle aggregation, a dose-dependent decrease in mitochondrial transmembrane potential, suppression of oxidative phosphorylation, an increase in H_2_O_2_ generation, induction of apoptosis in an ROS-mediated manner, suppression of the STAT3 activation pathway, autophagy, cell cycle arrest, inhibition of mtDNA replication, inhibition of the activity of complexes of the respiratory chain, influence on the balance between pro- and anti-apoptotic proteins and others (for more details, see [320,321]).

Despite the increasing number of anti-cancer drugs used for treatment of solid tumours, CDDP remains a widely used conventional therapy. However, the application of CDDP has numerous limitations, including drug resistance and off-target effects. To overcome these limitations, derivatives of this drug have been designed, synthetized and tested. Among them, monofunctional CDDP complexes represent a highly promising class of such derivatives, in which lonidamine (an inhibitor of mitochondrial hexokinase) is anchored to the CDDP centre for the selective de-energization of cancer cells. One such complex among the monofunctional Pt(II) complexes—monofunctional Pt(III) (MPL-III)—was reported to be more potent than CDDP in an MDA-MB-231 TNBC cell line, although it exhibited relatively low cytotoxicity towards breast epithelial cells. The MPL-III derivative was further shown to mainly accumulate in the mitochondria, where it induced detrimental changes to the mitochondrial ultrastructure, caused significant loss of the MMP, inhibited glycolysis and disrupted mitochondrial respiration. Consequently, MPL-III caused cell cycle arrest in the G0/G1 phase and mitochondria-mediated apoptosis involving caspase activation and cytochrome c release. At the molecular level, MPL-III was found to perturb DNA damage repair pathways, metabolic processes and transcription regulation [322]. Another way of overcoming CDDP resistance through mitochondrial targeting can be achieved with dinuclear Ir-CDDP complexes, in which an iridium(III) moiety is introduced to a terpyridyl CDDP derivative. The resulting compound, Ir-Pt, has exhibited a significant increase in mitochondrial accumulation and strong anti-tumour activity towards CDDP-resistant lung adenocarcinoma cells. This compound has been shown to severely damage mtDNA, disrupting the mitochondrial function, leading to loss of the MMP and depletion of ATP and resulting in cell death by necrosis [323,324].

Application of NPs provides a platform for more efficient ways to target mitochondria. For development of effective NP-based therapies, there is a substantial need to choose the right combination of mitochondria-penetrating peptides (MPPs) and cell-penetrating peptides (CPPs). To attack mtDNA with CDDP more efficiently using NPs, a hydrophobic mitochondria-targeting CDDP prodrug, Platine-M, was constructed through cycloaddition of an azide-Pt(IV) precursor to TPP^+^-bound azadibenzocyclooctyne. Platine-M loaded onto the surface of a biocompatible triblock polymeric NP accumulated effectively inside the mitochondrial matrix of chemoresistant ovarian carcinoma cells, which was accompanied by significantly better Platin-M activity than pure CDDP [325]. Another approach to target mitochondria of cancer cells using specific peptides as the basis for CDDP delivery involves tethering of *cis*-ammineplatinum(II) complex [Pt(succac)(NH_3_)_2_](NO_3_) to the N terminal end of the MPPs supplemented by unnatural amino acids, particularly D-arginine and L-cyclohexylalanine [326]. Interestingly, the linkage of Pt(IV) prodrug with oligonucleotide gold nanoparticle conjugates (DNA-AuNP) increased cytotoxicity compared to CDDP in human lung carcinoma A549, human prostate cancer PC3 and cervical cancer HeLa cell lines [327]. Short interfering RNAs, particularly silencing RNAs (siRNAs), conjugated with CPP (or other carriers) enable systemic delivery of siRNA into the cell, where it may induce sequence-specific posttranscriptional modifications in targeted transcripts crucially affecting further progress of cancer cells [328]. As an example, mesoporous silica nanoparticles (MSNs) loaded with siRNA targeting the protooncogene *BCL-2* along with a chemotherapeutic drug in the NP core showed a synergistic effect against chemoresistant TNBC [329]. Another dual drug conjugate composed of α-tocopheryl succinate (α-TOS) and CDDP, or, alternatively, doxorubicin or paclitaxel, blended with lipids and polyethylene glycol using a lipid-film hydration method was examined in the HeLa cell line. α-TOS elevated the effect of each individual drug, which was demonstrated by both nDNA damage and cytochrome c release from mitochondria [330]. In addition, targeting mitochondrial complex II of ETC with α-TOS conjugated to TPP^+^ increased the permeability of the resulting TOS-TPP^+^-obatoclax (BCL-2 inhibitor) NP in the outer mitochondrial membrane, resulting in apoptosis of TNBC cells [331]. Human breast cancer cell lines MCF-7 and T47D were successfully targeted by a covalent conjugate composed of nucleolin, the peptide targeting breast cancer cell receptor, and thiol groups containing α-lipoic acid when both were loaded onto an AuNP a stable receptor-specific peptide-AuNP was formed [332]. Eventually, mitochondria could be targeted with a dual drug conjugate composed of CDDP and the 3-bromopyruvate inhibitor that affects glycolytic enzyme hexokinase 2 within OXPHOS [325].

All the above examples prove that mitochondria indeed provide a suitable environment that may be therapeutically targeted to further improve cancer therapy. A number of studies have demonstrated how targeting different aspects of mitochondrial processes can successfully contribute to cancer treatment. However, the biggest challenge in the development of effective anti-cancer drugs is to create pharmacologic regulators respecting the multifaceted nature of the individual processes participating in tumorigenesis in each specific cancer type or subtype. Nevertheless, among the different therapeutic approaches, mitochondria-targeted DNA damaging agents still possess high potency and the ability to better evade therapy resistance mechanisms and off-target effects, combating the cancer cells through mechanisms that are distinct from those achieved by free drugs.

## 7. Concluding Remarks

During the last few decades, our view of mitochondria has shifted from the simple picture of the “powerhouse of the cell” to an organelle involved in complex processes driving the destiny of cells.

In cancer cell, mitochondria exhibit an enormous capacity to adapt and support tumour growth, adjusting the metabolism to increased energy demands, prioritizing OXPHOS or activating glycolysis as the main ATP source. Altered mtDNA profiles, including mtDNA copy number and mtDNA mutations, lead to modified mitochondrial bioenergetics and to pathologic mitochondria-to-nucleus signalling that accelerates tumorigenesis. In addition, mitoepigenetic modifications via methylation or acetylation affect retrograde communication and regulate gene expression of the nuclear-encoded oncogenic and tumour-suppressive genes. Likewise, dual regulatory roles are exerted by mitomiRs. MtDNA has increased susceptibility to ROS-induced mutations; therefore, it is not surprising that mitochondria possess several DNA repair systems. However, their repair capacity in tumours can be insufficient or defective, thus further contributing to cancer progression.

Any imbalance in mitochondrial homeostasis fundamentally contributes to neoplastic transformation, acceleration of the metastatic potential and therapy resistance development. In the search for new therapeutic strategies to bypass treatment resistance, development of new drugs based on cancer-specific organelle targeting seems to be a promising novel approach. In this context, searching for drugs focused on disrupting mitochondria-related processes in cancer cells represents an important approach for identifying novel selective systems and regimens with minimal side effects and high therapeutic benefits for cancer patients. Several new compounds with improved sensitivity have already been implemented, which also creates opportunities for new combination therapies.

## Figures and Tables

**Figure 1 ijms-23-07897-f001:**
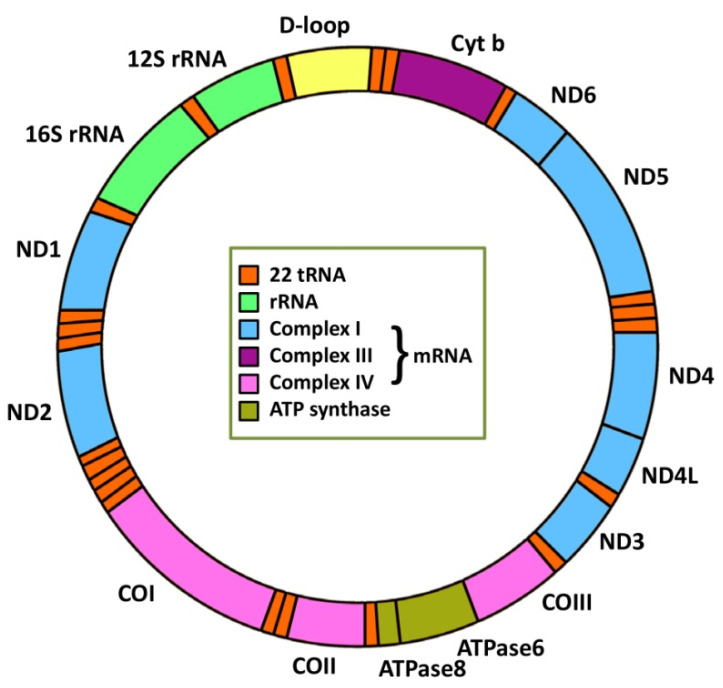
Mammalian mitochondrial DNA genome. mtDNA is a double-stranded circular DNA containing approximately 16.5 kb encoding 37 genes. These include 2 rRNAs, 22 tRNAs and 13 mRNAs. The D-loop region does not contain coding sequences. Cyt b: Cytochrome b; ND: NADH dehydrogenase; CO: cytochrome c oxidase; ATPase: ATP synthase.

**Figure 2 ijms-23-07897-f002:**
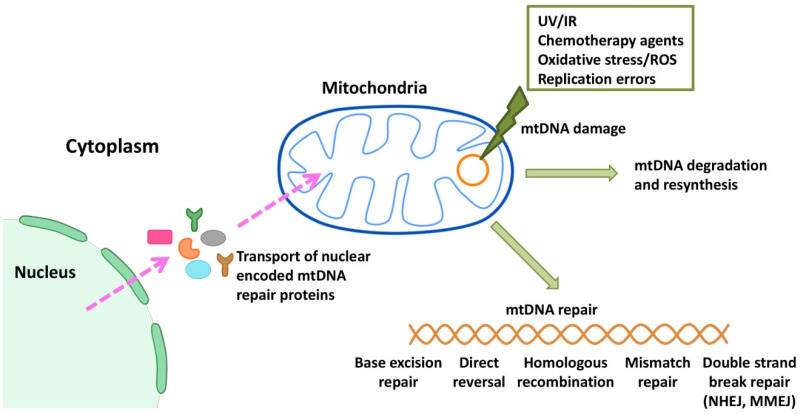
mtDNA damage repair in humans. Different stressors can cause mtDNA damage. If it is unrepairable, the mtDNA is degraded and newly synthetized. Proteins engaged in mtDNA repair are nuclear-encoded and transported to mitochondria. Base excision repair is the main mtDNA repair pathway. Other pathways show minor representation, lack full functional validation in humans or are only partially described. UV: ultraviolet light; IR: ionizing radiation; NHEJ: non-homologous end joining; MMEJ: microhomology-mediated end joining.

**Figure 3 ijms-23-07897-f003:**
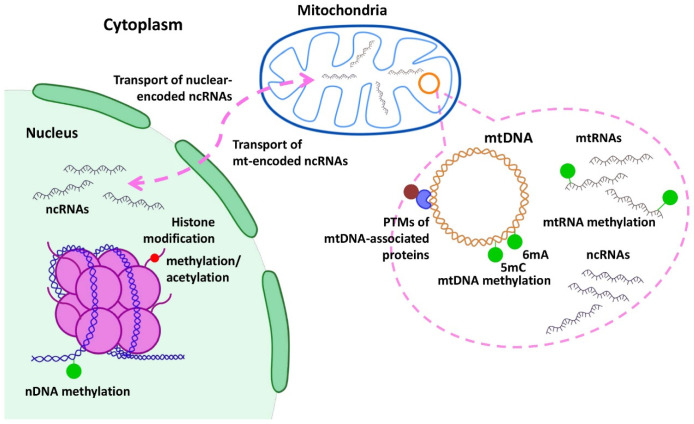
Overview of mitochondrial epigenetics. Modulation of mitochondrial gene expression is regulated by mtDNA/mtRNA methylation, non-coding RNAs and PTMs of mtDNA-associated proteins. mtDNA methylation refers to 5mC and 6mA methylation. ncRNAs, both nuclear- and mitochondria-encoded, regulate transcription, translation and post-translation via interaction with DNAs, RNAs and proteins. ncRNAs also take part in mitochondria-to-nucleus communication. PTMs of mtDNA-associated proteins include acetylation, phosphorylation, methylation, SUMOylation, ubiquitination and PARylation. ncRNAs: non-coding RNAs; PTMs: post-translational modifications; 5mC: 5-methylcytosine; 6mA: N^6^-methyladenine.

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
