# Peer review of "Mitochondrial Genetic and Epigenetic Regulations in Cancer: Therapeutic Potential"

_ijms, 2022, doi:10.3390/ijms23147897_

Round 1

Reviewer 1 Report

This is a fairly high-quality review presented by Wagner and colleagues, describing changes in the functioning of mitochondria in cancer, which are caused by genetic rearrangements. I have some comments on its content:

1. Line 67. Mitochondria – originally proteobacteria swallowed by a pre-eukaryotic cell.

The authors shouldn't be so blunt about it. This is still just a guess.

2. In addition to the strong dependence on glycolysis, many tumours or subpopulations of cells in tumours rely on OXPHOS for bioenergetics (Caro et al. Cancer Cell 2012; Goto et al., Cancer Invest. 2014; Haq, R. et al. Cancer Cell 2013) and biosynthetic processes (Birsoy et al. Cell 2015; Sullivan et al, Cell 2015), which provide the basis for the therapeutic targeting of mitochondria. Furthermore, an increased dependence of tumour cells on OXPHOS was observed in the advanced stages of the disease (Faubert et al, 2020 Science).

3. Refs 5 and 6 are quite old, they need to be updated. There are more recent reviews.

4. In part 6, the purpose of using lipophilic cations should be noted. Delocalized lipophilic cations are used to deliver compounds with anticancer activity, but low bioavailability (causing their systemic toxicity). Conjugation with with mitochondria-targeted lipophilic cations (TPP+, F16, rhodamine B) solves this problem in some cases. I also recommend considering the contributions of René Csuk and Anna Spivak's groups to the development of mitocans (this acronym should also be noted) based on selective cytotoxic triterpenoids.

Author Response

We are very thankful to the Reviewer 1 for all constructive comments, suggestions and recommendations.  Please, find our point-by-point response to the reviewer’s comments below:

Comments and Suggestions for Authors

This is a fairly high-quality review presented by Wagner and colleagues, describing changes in the functioning of mitochondria in cancer, which are caused by genetic rearrangements. I have some comments on its content:

  1. Line 67. Mitochondria – originally proteobacteria swallowed by a pre-eukaryotic cell.

The authors shouldn't be so blunt about it. This is still just a guess.

Response: Thank you for pointing this out. The sentence was reformulated as following:  “Mitochondria – arose in evolution presumably as a proteobacteria swallowed by a pre-eukaryotic cell,…”.

  1. In addition to the strong dependence on glycolysis, many tumours or subpopulations of cells in tumours rely on OXPHOS for bioenergetics (Caro et al. Cancer Cell 2012; Goto et al., Cancer Invest. 2014; Haq, R. et al. Cancer Cell 2013) and biosynthetic processes (Birsoy et al. Cell 2015; Sullivan et al, Cell 2015), which provide the basis for the therapeutic targeting of mitochondria. Furthermore, an increased dependence of tumour cells on OXPHOS was observed in the advanced stages of the disease (Faubert et al, 2020 Science).

Response: Thank you for suggestion of this upgrading and supporting information. Suggested sentences and references have been incorporated into the text of the revised version of our manuscript.

  1. Refs 5 and 6 are quite old, they need to be updated. There are more recent reviews.

Response: Thank you for pointing this out. We have replaced original Refs 5 and 6 with more recent reviews:

[5]       J.B. Spinelli, M.C. Haigis, The multifaceted contributions of mitochondria to cellular metabolism, Nat. Cell Biol. 20 (2018) 745–754. https://doi.org/10.1038/s41556-018-0124-1.

[6]      Y. Ma, L. Wang, R. Jia, The role of mitochondrial dynamics in human cancers., Am. J. Cancer Res. 10 (2020) 1278–1293.

  1. In part 6, the purpose of using lipophilic cations should be noted. Delocalized lipophilic cations are used to deliver compounds with anticancer activity, but low bioavailability (causing their systemic toxicity). Conjugation with with mitochondria-targeted lipophilic cations (TPP+, F16, rhodamine B) solves this problem in some cases. I also recommend considering the contributions of René Csuk and Anna Spivak's groups to the development of mitocans (this acronym should also be noted) based on selective cytotoxic triterpenoids.

Response: Thank you for your suggestions and recommendations definitely improving this part of our manuscript. We have revised the whole Section 6 accordingly. Please see the revised version of the manuscript.  

In addition, we made few corrections of the spelling.

Reviewer 2 Report

In their paper entitled "Mitochondrial genetic and epigenetic regulations in cancer: therapeutic prospective," the authors summarized the current state of knowledge in mitochondrial genetics as it relates to cancer cells. Particular attention was paid to epigenetic regulation of mitochondrial genes and miRNA regulation. The last two sections deal with mitochondrial targeting and therapeutic perspectives. The review is very well written in detail and supported with current literature. The present form is suitable for publication.

Author Response

We kindly thank the Reviewer 2 for his time and reviewing our manuscript.

Round 2

Reviewer 1 Report

The authors have significantly improved the presentation of the work. I have no more comments.

Author Response

We kindly thank Reviewer 1 for reviewing the revised version of our manuscript.